# Adapter to facilitate Foundation Model Communication for DLO Instance Segmentation

**Omkar Joglekar**[*], **Shir Kozlovsky**[*]
Bosch Center for Artificial Intelligence
Haifa, Israel
`{joo1tv, kos1tv}@bosch.com`

**Dotan Di Castro**
Bosch Center for Artificial Intelligence
Haifa, Israel
`did1tv@bosch.com`

**Editors:** Marco Fumero, Clementine Domine, Zorah Lähner, Donato Crisostomi, Luca Moschella, Kimberly Stachenfeld

## Abstract

Classical methods in Digital Communication rely on mixing transmitted signals with carrier frequencies to eliminate signal distortion through noisy channels. Drawing inspiration from these techniques, we present an adapter network that enables CLIPSeg, a text-conditioned semantic segmentation model, to communicate point prompts to the Segment Anything Model (SAM) in the positional embedding space. We showcase our technique on the complex task of Deformable Linear Object (DLO) Instance Segmentation. Our method combines the strong zero-shot generalization capability of SAM and user-friendliness of CLIPSeg to exceed the SOTA performance in DLO Instance Segmentation in terms of DICE Score, while training only $0.7\%$ of the model parameters.

## 1 Introduction

Deformable Linear Objects (DLOs), encompassing cables, wires, ropes, and elastic tubes, are commonly found in domestic and industrial settings Keipour et al. (2022); Sanchez et al. (2018). Despite their widespread presence in these environments, DLOs present significant challenges to automated robotic systems, especially in perception and manipulation, as discussed by Cop et al. (2021). In terms of perception, the difficulty arises from the absence of distinct shapes, colors, textures, and prominent features, which are essential factors for precise object recognition.

Over the past few years, there has been a notable emergence of approaches customized for DLOs. State-Of-The-Art (SOTA) instance segmentation methods such as mBEST Choi et al. (2023) and RT-DLO Caporali et al. (2023) use unique approaches influenced by bending energy or by classic concepts from graph topology, to segment these challenging objects accurately. However, none of these methods excel in handling real and complex scenarios, nor do they incorporate prompt-based control functionality for segmentation, such as text prompts, which enhance user accessibility.

In the domain of purely computer vision-based approaches, the Segment Anything Model (SAM; Kirillov et al. (2023)) is one of the most notable segmentation models in recent years. As a foundation model, it showcases remarkable generalization capabilities across various downstream segmentation tasks, using smart prompt engineering Bomasani and Others (2021). However, SAM's utility is limited to manual, counter-intuitive prompts in the form of points, masks, or bounding boxes, with basic, proof-of-concept text prompting. On the other hand, in the domain of deep vision-language

---

[*]Both authors contributed equally to this work, the order was finalized alphabetically.

fusion, CLIPSeg Lüddecke and Ecker (2021) presents a text-promptable semantic segmentation model. However, this model does not extend its capabilities to instance segmentation.

To address these challenges, we present a novel adapter model that harnesses the strengths of these Foundation Models (FMs), as illustrated in Fig. 1. In this adapter we introduce a novel *sampler attention* layer, that samples point prompt embeddings from the same 2D frequency space that SAM is trained to process, conditioned on text-dependent attention heatmaps generated by CLIPSeg.

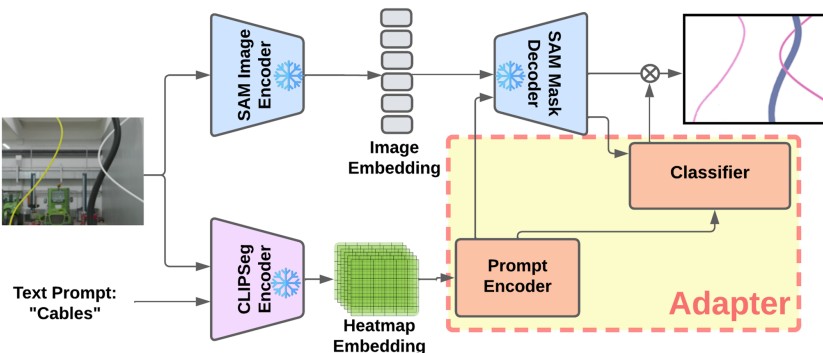

Figure 1: Overview of the full pipeline - blocks in red represent our additions

With our approach, users can achieve precise one-shot instance segmentation of DLOs by merely supplying an image and a text prompt. The main contributions of our work can be summarized as follows:

1. A novel **prompt encoding network** that translates semantic information from text prompts into point prompt embeddings that SAM can interpret.
2. The implementation of a **classifier network** designed to eliminate redundant and low-quality masks produced by SAM, ensuring the reliability of the segmentation output.
3. The creation of a comprehensive, **CAD-generated dataset** tailored to DLOs, featuring approximately 30,000 high-resolution images of industrial cables. This dataset is a valuable resource for training and validating our model.

Our model achieves an $mIoU = 91.21\%$ on our custom dataset, significantly outperforming RT-DLO's $mIoU = 50.13\%$. Additionally, our model demonstrates exceptional zero-shot transfer capabilities to datasets used in RT-DLO Caporali et al. (2023) and mBEST Choi et al. (2023), exceeding the SOTA DICE score in DLO instance segmentation.

## 2 Related Work

### 2.1 DLOs Instance Segmentation and Manipulation

Several joint DLO segmentation and manipulation algorithms such as Viswanath et al. (2023), Chi and Berenson (2019), Chi et al. (2022), Grannen et al. (2020) and Nair et al. (2017) perform manipulation and segmentation simultaneously and on real robot systems. These novel techniques however do not report comparable baselines scores such as mIoU or DICE score for independent comparison of segmentation baselines.

DLO detection algorithms employ various methodologies to address specific challenges. Zanella et al. (2021) introduced the first CNN-based approach for DLO semantic segmentation. Yan et al. (2019) and Keipour et al. (2022) pioneered data-driven techniques for DLO instance segmentation, using neural networks to reconstruct DLO topology and fitting curvatures and distances for continuous DLOs, respectively. These methods were limited to single DLO detection.

*Ariadne* De Gregorio et al. (2018) and *Ariadne+* Caporali et al. (2022) segment DLOs into superpixels Achanta et al., then use path traversal to generate instance masks. *Ariadne+* uses DeepLabV3+ Chen et al. (2018) to extract a semantic mask before superpixel processing, overcoming *Ariadne's* limitations. Both methods struggle with numerous hyperparameters. FASTDLO Caporali et al. (2022)

and RT-DLO Caporali et al. (2023) use skeletonization of semantic masks instead of superpixels, enhancing speed. FASTDLO processes the skeletonized mask with path traversal and a neural network for intersections, while RT-DLO uses graph node sampling from DLO centerlines, followed by topological reasoning for edge selection and path determination. Despite near-real-time performance, they are sensitive to noise from centerline sampling and require scene-dependent hyperparameter tuning. mBEST Choi et al. (2023) fits a spline to minimize bending energy during skeleton traversal, showing impressive results but limited to fewer than two intersections between DLOs and relies on a manually tuned threshold.

All these approaches, including SOTA baselines RT-DLO Caporali et al. (2023) and mBEST Choi et al. (2023), are sensitive to occlusions and cables near image edges, common in industrial settings. They also struggle with dataset transfer and specific operating conditions. Our approach addresses these issues using a diverse dataset and the generalization power of foundation models. To our knowledge, this is the first end-to-end, computer vision-based method combining DLO instance segmentation with text prompts for a user-friendly solution.

## 2.2 Text-conditioned Instance Segmentation

SAM Kirillov et al. (2023) is a powerful foundation model with remarkable performance on various downstream tasks, available in three variants (ViT-B, ViT-L, ViT-H) with increasing size and performance. SAM excels in segmenting DLOs, leveraging points, boxes, or masks for segmentation prompts. It outperforms other promptable segmentation models Chen et al.; Ding et al.; Liu et al. (2022a,b) in generalizability and quality. However, SAM's text-based prompting still requires manual point prompts for high-quality segmentation. Box prompts are ineffective for one-shot instance segmentation as they can encompass multiple DLOs, and single-point prompts struggle in complex scenarios, making manual input inefficient for industrial automation (Fig. 2).

Methods like Grounded-SAM Ren et al. (2024) and open-source repositories like Panoptic Segment Anything Segments AI (2023) use frozen VLMs such as GroundingDINO Liu et al. (2023) for prompting SAM, relying heavily on bounding boxes, which are unsuitable for DLO segmentation.

CLIPSeg Lüddecke and Ecker (2021) is a text-guided semantic segmentation model with an embedding space representing spatial attention heatmaps from text prompts. We utilize SAM's segmentation and point prompt capabilities, introducing an alternate prompt encoder that converts text prompts into point-prompt embeddings. Our model performs instance segmentation in one forward pass without sequential mask refinement and includes a network to filter out duplicate and low-quality masks.

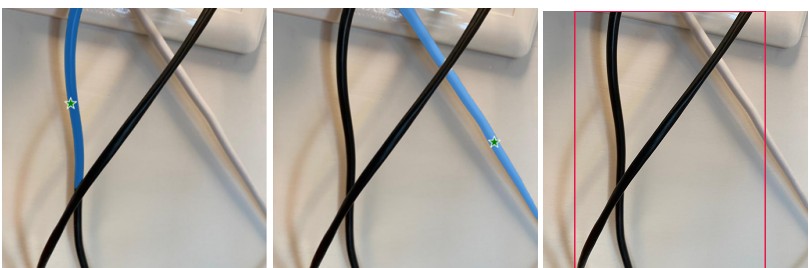

(a) An example of unsuccessful segmentation with a single point.
(b) An example of successful segmentation with a single point
(c) An example of box prompt for DLO segmentation.

Figure 2: Issues with using SAM out-of-the-box

## 2.3 Adapters

Adapters are methods that incorporate additional trainable parameters into a frozen pretrained model, enhancing its ability to learn and perform downstream tasks. They are part of the (Parameter-Efficient Fine-Tuning) PEFT family Xu et al. (2023); Yu et al. (2023) and can interact with FMs as trainable blocks inserted into the FM architecture Houlsby et al. (2019); Lin et al. (2020) or come as an additional component to an FM Pfeiffer et al. (2021). This approach was initially introduced in the Natural Language Processing (NLP) domain Houlsby et al. (2019) as a way to facilitate Parameter-

Efficient Transfer Learning (PETL)Yu et al. (2023). Due to its exceptional efficacy, this technique has also found successful application in the field of computer vision (CV) Xin et al. (2024).

## 2.4 Positional Encoding

SAM uses positional encodings that are derived from Fourier features, inspired by Tancik et al. (2020), to encode the point prompts. These Fourier features are calculated based on a randomly initialized frequency matrix. This can be viewed as transmitting 2D point vectors in the 2D Fourier domain, where each element of the embedding vector denotes the amplitude of a particular frequency from this frequency matrix. This allows us to apply classical techniques from Digital Communication such as carrier frequency synchronization Ling (2017), to communicate point prompts using the same carrier frequencies. In this method we enforce this carrier synchronization by embedding our point prompts using the same frequency matrix used by SAM.

## 2.5 Latent Space Communication

Recently, there has been growing agreement that good networks learn similar representations across a variety of architectures, tasks and domains Morcos et al. (2018); Barannikov et al. (2022). In authors Lenc and Vedaldi (2015), introduce trainable stitching layers that allow swapping parts of different networks. This line of research induced a new field of research called Latent Space Communication (LSC), a term coined by Moschella et al. (2023), where the authors studied how relative latent representations can be transferred across model architectures, tasks, datasets and modalities, to allow zero-shot model stitching. Other works such as Lähner and Moeller (2024) prove that linear transformations can be used to perform LSC without access to prior knowledge. Cannistraci et al. (2023) propose a versatile method to infuse latent space invariance without prior knowledge about its optimality. Our method draws inspiration from these prior works to perform LSC, using a trainable adapter network to perform model stitching. We assume that we have access to a small amount of prior knowledge, such as the frequency matrix of SAM's positional encoding.

# 3 Methods

In this section, we describe the core method behind our approach.

## 3.1 The Model Components

The adapter model we propose consists of 2 main networks (as depicted in Fig. 1):

1. **Prompt encoder network** - This network samples batches of point prompt embeddings in the same frequency space that SAM is trained to process, using CLIPSeg's embedding space. It can be controlled using 2 hyperparameters: $N$ (the number of prompt batches) and $N_p$ (the number of points in each batch).

2. **Classifier network** - A *binary* classifier network that labels whether a particular mask from the generated $N$ masks should appear in the final instance segmentation.

Based on the discussion of adapters in Sec. 2.3, we freeze both FMs, to achieve high performance at a lower computational cost.

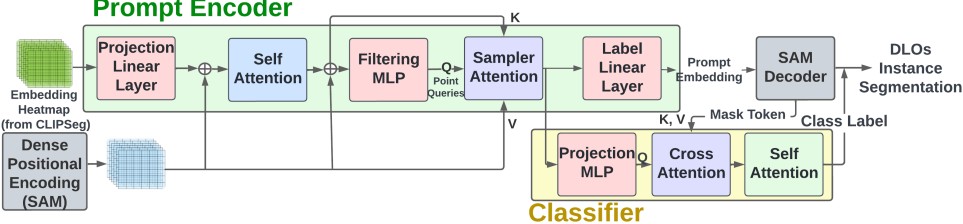

Figure 3: The Adapter: **top** - the prompt encoder network is outlined (indicated by the green box); **bottom** - the classifier network architecture is detailed (represented by the yellow box).

### 3.1.1 Prompt encoder network

We propose the prompt encoder network as the trainable module to bridge the gap between a frozen CLIPSeg and frozen SAM. It extracts a fixed number of point prompt embedding vectors from CLIPSeg's embedding space that SAM's mask decoder can accurately decipher. As outlined in Fig. 3, the prompt encoder network takes as an input the upsampled attention embedding obtained from CLIPSeg, that embeds a semantic mask that spatially aligns with the input image and is conditioned on text. This embedding is enhanced with Dense Positional Encoding (DPE) to ensure that the self-attention layer can access crucial geometric information. To generate our DPE, we use an identical frequency matrix as the SAM. This ensures that every component within each DPE vector conveys consistent information aligned with what SAM's decoder has been trained to interpret. These enhanced patch embeddings undergo a single self attention operation to learn inter-patch correlations.

The following MLP filters out viable query patches $N \times N_p$. These queries are used to score the space of the DPE, in the sampler attention. The output of this block is $N \times N_p$ point embeddings that SAM's decoder is trained to process. A single linear layer is applied to this output to categorize each chosen point as foreground/background/no-point. This layer aims to mimic the SAM labeling protocol, which employs a standard affine transformation, where a learned embedding is added to each point embedding corresponding to its category. Finally, this output is reshaped to the size $(N, N_p, 256)$ as a batch of $N$ prompts, each containing $N_p$ points, where 256 is the embedding dimension for point prompts, in SAM.

Following the original implementation of transformers Vaswani et al. (2023), queries are reintroduced to the attention outputs and layer normalization Ba et al. (2016) is applied after each attention layer. Additionally, DPE is added to each attention output to ensure that crucial spatial information of the image is propagated throughout the network.

It is important to note that our DPE, like SAM, takes inspiration from Fourier features Tancik et al. (2020) alongside classical techniques from digital communication Ling (2017), using the same frequency matrix as SAM instead of a randomly generated one. All the attention blocks shown in the diagram are *single layers* of attention.

### 3.1.2 Mask classification network

In previous works, such as DETR Carion et al. (2020) and MaskFormer Cheng et al. (2021), the authors train a classifier network along with the box regression model to classify which object is contained within the box. In later works such as Deformable DETR Zhu et al. (2020) and DETR3D Wang et al. (2021), the authors show the advantages of using anchor point-based queries rather than randomly initialized ones. We compare the impact of randomly initialized object queries and anchor point-based object queries in Appendix C. All of these works introduced an additional *no-object class* to filter out duplicate or erroneous boxes/masks during prediction. Inspired by them, we developed a binary classifier network for this purpose.

This binary classifier model comprises a projection MLP, one cross-attention block, one self-attention block, and a classifier MLP, as illustrated in Fig. 3. A detailed motivation for the network architecture is discussed in Appendix C.

First, the sampled point embeddings (from our prompt encoder) are transformed using an MLP. Second, a cross-attention block operates on these transformed embeddings generated by our model, which encodes text-conditioned information, and on the mask tokens produced by SAM, which encapsulate submask-related details. This interaction results in an embedding that fuses both types of information for each generated submask. Subsequently, the queries are combined with the output to reintroduce textual information to these *classifier tokens*. These classifier tokens then undergo a self-attention layer followed by the MLP to produce binary classifications.

### 3.2 Training protocol

The full architectural framework of our model, illustrated in Fig. 1, accepts a single RGB image and a text prompt as input, delivering an instance segmentation mask for each DLO in the image. In our setup, we keep a constant text-prompt "cables" for the whole training session. Inspired by the work of Cheng et al. (2021), we employ a bipartite matching algorithm Kuhn (1955) to establish the optimal correspondence between the generated masks and ground-truth submasks to train the prompt encoder.

We use a combination of the focal loss Lin et al. (2017) and the DICE loss Milletari et al. (2016) in the ratio of $20 : 1$, as recommended by Kirillov et al. (2023).

The binary classification labels are extracted from the output of the bipartite matching algorithm. If a mask successfully matches, then it is labeled as 1; else, it is labeled as 0. We use the binary cross-entropy loss to train this network. To balance the distribution of 0s and 1s in the dataset, we introduce a weight for positive labels that we set to 3.

The most time-intensive step within the SAM workflow involves generating the image embedding. However, once this image embedding is created, it can be used repeatedly to produce multiple submasks as needed. This approach significantly speeds up the process and can be executed in a single step. To enhance training efficiency using this method, we set the training batch size to just 1. This configuration enables us to form batches of prompts, with each DLO in the scene associated with an individual prompt. We place a cap of $N = 11$ prompts for each image and limit the number of points within each prompt to $N_p = 3$. Specific details about the training process can be found in Appendix A.

## 4 Experiments

### 4.1 The cables dataset

The dataset used for training, testing and validation was generated using Blender[2], a 3D rendering software. It comprises images showing various cables of various thicknesses, shapes, sizes, and colors within an industrial context, exemplified in Figure 10. The dataset consists of 22k training images, along with 3k images, each designated for validation and testing. Each image has a resolution of 1920x1080, and each DLO within an image is accompanied by a separate mask, referred to as a submask in this work. This pioneering DLO-specific dataset encompasses an extensive range of unique scenarios and cable variations. We anticipate that granting access to this comprehensive dataset will significantly advance numerous applications in the field of DLO perception within industrial environments. For more details and sample images from the dataset, refer to the Appendix B. This dataset is available on Zenodo[3], with the title "DLO Instance Segmentation dataset generated by Blender" (Joglekar et al. (2023)).

### 4.2 Baseline experiments

While designing the model and configuring the best hyperparameters, we conducted multiple experiments. The initial experiment involved selecting the number of points per prompt, denoted as $N_p$. In the SAM framework, each point can be labeled as either foreground or background. The SAM paper Kirillov et al. (2023) explored various values for $N_p$ and documented the resulting increase in Intersection over Union (IoU), observing diminishing returns after 3 points. For one-shot applications, we reevaluated this parameter using $N_p = 2, 3, 4$ and concluded that $N_p = 3$ is optimal, adopting this value for subsequent experiments.

The second experiment focused on SAM's capacity to generate three masks per prompt to handle ambiguous prompts. We compared this multi-mask output with the single-mask output in terms of mean IoU (mIoU). Our findings indicated that multi-mask output led to only minor improvements but significantly higher memory utilization. Consequently, we disabled the multi-mask functionality for all subsequent experiments involving SAM.

### 4.3 Quantitative experiments

In these experiments, our primary objective is to compare our model's DLO instance segmentation results with SOTA baselines and assess the model's limitations using the *Oracle* method Kirillov et al. (2023). This method involves removing the classifier network during testing and using bipartite matching to filter out duplicate and low-quality masks, allowing independent evaluation of our prompt encoder and classifier networks. During Oracle tests, our model has access to ground-truth annotations.

---

[2] https://www.blender.org/
[3] Zenodo

We test the overall quality of generated instance masks, zero-shot generalization capabilities, and the effect of image augmentation on generalization. In Tab. 2, demonstrating our models' zero-shot transfer capabilities to the testing datasets discussed in mBEST Choi et al. (2023).

## 5 Results

### 5.1 Quantitative results

We summarize the comparison of the results of different configurations of our model in our test dataset in Tab. 1. As a reference, we tested the RT-DLO algorithm Caporali et al. (2023) on our test dataset. It achieved $mIoU = 50.13\%$. Oracle tests show the upper bound of the performance that our method can achieve. These limitations are discussed in depth in the appendix D.

| Model Configuration | Test data performance mIoU [%] | DICE [%] | Augmentation | Oracle |
|---|---|---|---|---|
| **A** (Aug only) | 90.64 | 99.78 | Y | N |
| **A+O** (Aug+Oracle) | 92.10 | 99.80 | Y | Y |
| **B+O** (Base+Oracle) | **92.51** | **99.82** | N | Y |
| **B** (Base) | 91.21 | 99.77 | N | N |

Table 1: Comparison of our model configurations (values in bold signify the best performance).

Table 2 shows the DICE scores of our model compared to current SOTA baselines: Ariadne+ Caporali et al. (2022), FASTDLO Caporali et al. (2022), RT-DLO Caporali et al. (2023) and mBEST Choi et al. (2023). The datasets C1, C2, C3 are published by Caporali et al. (2023), while the datasets S1, S2, S3 are published by Choi et al. (2023).

In the *no Oracle* setting, we observe that the model generalizes better if it is trained using augmentations in the dataset, as expected. Our model exhibits strong zero-shot generalization to all the datasets. Specifically for C1, C2, and C3, our model exceeds SOTA performance even in the base configuration. Furthermore, in the case of Oracle configuration, we see that our proposed method outperforms all SOTA baselines on all datasets, showing that the mask classifier network is a bottleneck in our full model.

| | | | | DICE[%] | | | | |
|---|---|---|---|---|---|---|---|---|
| | | | | | **Our Model** | | | |
| Dataset | Ariadne+ | FASTDLO | RT-DLO | mBEST | A | A+O | B+O | B |
| C1 | 88.30 | 89.82 | 90.31 | 91.08 | 97.03 | 98.85 | **98.86** | 94.55 |
| C2 | 91.03 | 91.45 | 91.10 | 92.17 | 96.91 | 98.69 | **98.70** | 96.12 |
| C3 | 86.13 | 86.55 | 87.27 | 89.69 | 97.13 | 98.81 | **98.90** | 90.26 |
| S1 | 97.24 | 87.91 | 96.72 | 98.21 | 97.36 | 98.43 | **98.60** | 93.08 |
| S2 | 96.81 | 88.92 | 94.91 | 97.10 | 97.71 | **98.54** | 98.42 | 97.01 |
| S3 | 96.28 | 90.24 | 94.12 | 96.98 | 96.69 | **98.79** | 98.58 | 95.83 |

Table 2: DICE score comparison for out-of-dataset, zero-shot transfer. Values in bold signify the best performance

To assess generalizability across various text prompts, we tested our model on prompts like "wires" and "cords," in addition to our training prompt, "cables," which specifically refers to DLOs. Table 3 showcases the performance of these prompts when assessed on the same model using Oracle. Remarkably, our model demonstrates the ability to effectively generalize across different prompts zero-shot, producing results comparable to the base prompt, "cables."

### 5.2 Qualitative results

This section presents the actual instance masks generated by our model. We used the configuration trained on augmentations (A) to generate all these results (without Oracle).

| Text Prompt | mIoU [%] | DICE [%] |
|---|---|---|
| cables (baseline) | 92.51 | 99.82 |
| wires | 89.6 | 99.77 |
| cords | 89.3 | 99.64 |

Table 3: Generalizability across text prompts: Demonstrates results comparable to the base prompt "cables" through zero-shot transfer.

Fig. 4 presents the instance masks generated by our model in various complex scenarios, along with the results of RT-DLO on the same image set for comparison. These scenarios encompass occlusions, DLOs with identical colors, varying thicknesses, small DLOs in the corners, a high density of cables in a single image, and real-world scenarios. Our model shows impressive performance across all of these scenarios.

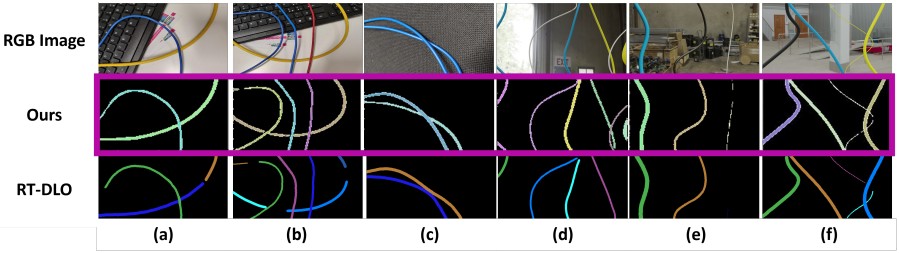

Figure 4: Qualitative comparison in specific scenarios. Each scenario demonstrates the following: (a) and (b) real images, (c) identical colors, (d) a high density of cables in a single image, and (e) and (f) small DLOs at the edge of the image with varying thicknesses.

In Fig. 5, we compare the generation of instance masks between our model and the current SOTA baselines on an external dataset. Our model demonstrates remarkable zero-shot transfer capabilities when applied to data distributions different from training data.

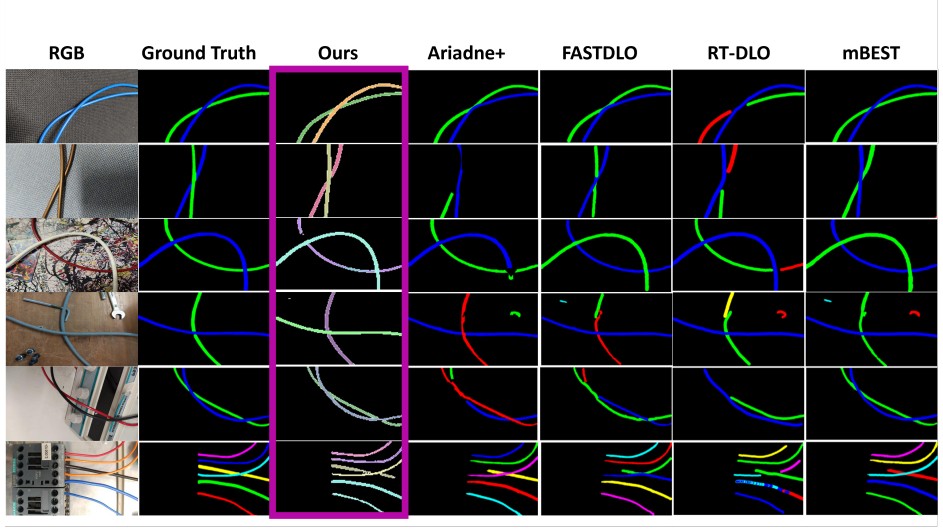

Figure 5: A qualitative comparison of our model vs. the SOTA baselines

Finally, we test our model on real images taken in our lab using a smartphone camera. The final output masks are shown in Fig. 6. We see that our model generalizes to real-world, complex scenarios of zero-shot. However, because of an imperfect classifier, it misses some submasks in certain complex situations. We report more generated instance masks in Appendix E.

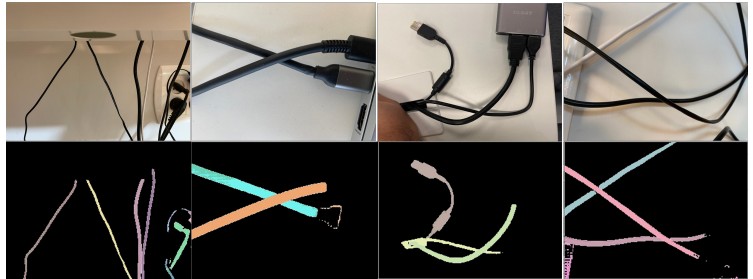

Figure 6: Qualitative results on real-world, complex scenarios zero-shot

## 6 Limitations

Our approach merges the capabilities of two significant foundational models (FMs): SAM and CLIPSeg, which are built on computationally heavy architectures of ViT-LDosovitskiy et al. (2020) and CLIPRadford et al. (2021), respectively. This integration leads to a computational time of approximately $330[ms]$ on a single NVIDIA 2080Ti GPU. Furthermore, challenges remain in segmenting cables with multiple self-loops, particularly in the S2 and S3 datasets, where our current setup does not utilize Oracle. We hypothesize that this can be attributed to the lack of similar images in our training dataset. However, this issue can potentially be mitigated through a few-shot fine-tuning on these specific datasets. Other limitations associated with the FMs used are discussed in Appendix D.

## 7 Conclusions and future work

In this paper, we introduce a lightweight adapter model to enhance the robust segmentation capabilities of SAM by leveraging the text-conditioned semantic segmentation of CLIPSeg. This method performs embedding space alignment by drawing inspiration from classical techniques in digital communication. We also present an innovative method for automatically filtering out duplicate and low-quality instance masks produced by SAM, addressing its primary limitations. These advances provide a more accessible approach for users to perform one-shot instance segmentation tasks with simple prompts.

Future work will extend our adapter model to one-shot instance segmentation of conventional rigid objects. Currently, our model's performance is limited by our classifier network. Enhancing the classifier network is a critical avenue for ongoing research to fully unlock the model's potential for broader segmentation tasks.

Our adapter significantly improves SAM's performance on the challenging task of DLO instance segmentation, achieving SOTA DICE scores (Sec. 5.1) with zero-shot transfer capabilities. Additionally, we contribute a comprehensive dataset, detailed in Sec. 4.1, to foster further research within this domain.

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

# A    Training details

In our method, we chose the ViT-L/16-based model of SAM (Kirillov et al., 2023) to attempt to balance the speed-accuracy trade-off. We observed that this model gave high-quality segmentation masks while being $\sim 2\times$ smaller compared to ViT-H/16 in terms of the number of parameters. On the other hand, for the CLIPSeg (Lüddecke and Ecker, 2021) model, we used the ViT-B/16 based model, with $reduce\_dim = 64$. Throughout our training process, we freeze the weights and biases of both foundational models, SAM and CLIPSeg.

Throughout the training phase, our text prompt remains "cables," with the aim of obtaining instance segmentation for *all* the cables within the image. During training, we conduct augmentations applied to the input images. These augmentations encompass random grayscale conversion, color jitter, and patch-based and global Gaussian blurring and noise. In terms of computing, our model is trained using 2 NVIDIA A5000 GPUs, each equipped with 32 GB of memory. All testing is carried out on a single NVIDIA RTX 2080 GPU with 16 GB of memory.

Our training procedure employs a learning rate warm-up spanning 5 epochs, followed by a cosine decay. The peak learning rate is set to $lr = 0.0008$, in line with recommendations from Kirillov et al. (2023). We employ the default *AdamW* optimizer from Paszke et al. (2019) with a weight decay of $0.01$. The convergence graphs and learning rate profile of all the models can be seen in Figure 7. Figure 8 displays the binary classification accuracy and mIoU computed on the validation dataset during training. No smoothing was applied in creating the plots.

In the training process, we apply augmentations such as blurring, color jitter and random grayscale, to enhance generalizability across various DLO colors.

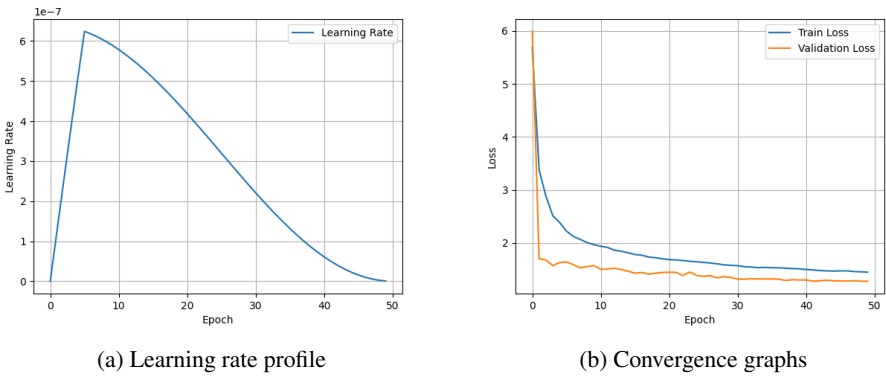

(a) Learning rate profile     (b) Convergence graphs

Figure 7: Learning rate profile and convergence graphs

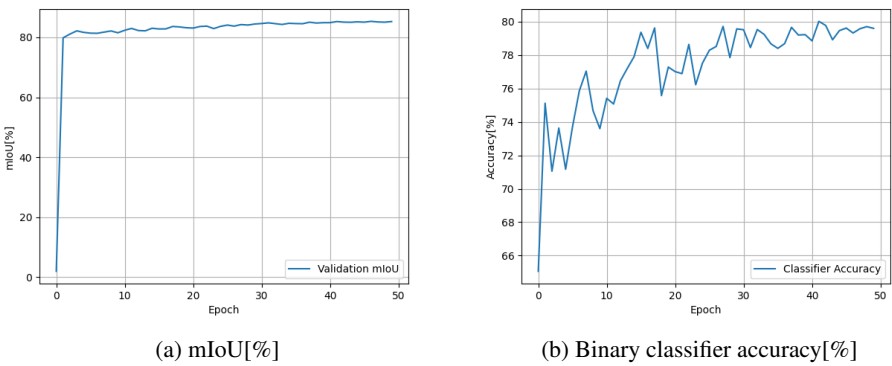

(a) mIoU[%]     (b) Binary classifier accuracy[%]

Figure 8: Binary classification accuracy and mIoU

A brief summary of all the *hyperparameters* can be seen the Table 4

| Hyperparameter | Value |
|---|---|
| Number of epochs | 50 |
| Max learning rate | $8 \times 10^{-4}$ |
| Learning rate warmup | 5 (epochs) |
| Optimizer | *AdamW* |
| Optimizer weight decay | 0.01 |
| Batch size | 1 |
| Attention dropout | 0.5 |
| Number of prompts per batch ($N$) | 11 |
| Number of points per prompt ($N_p$) | 3 |
| Number of attention heads (for all models) | 8 |
| Embedding dimension | 256 |
| SAM model type | *ViT-L/16* (frozen) |
| CLIPSeg model type | *ViT-B/16* (frozen) |
| Focal loss weight | 20 |
| DICE loss weight | 1 |
| Positive label weight (binary cross-entropy) | 3 |
| Classifier MLP activation | *ReLU* |
| Prompt encoder MLP activation | *GELU* |
| Train dataset size | 20038 |
| Validation dataset size | 3220 |
| Test dataset size | 3233 |
| Image size | $(1920 \times 1080)$ |
| Total number of parameters (including foundation models) | 466M |
| Trainable parameters | 3.3M |

Table 4: Hyperparameters

## B   Generated dataset

The dataset we presented contains 20038 train images, 3220 validation images, and 3233 test images. Each image we provide is accompanied by its corresponding semantic mask and binary submasks for each DLO in the image. The images are located in {train,test,val}/RGB, and named as train/RGB/00000_0001.png, train/RGB/00001_0001.png, and so on. In the {train, test, val}/Masks folder, we have a sub-folder containing the binary submasks for each corresponding RGB image. For example, train/Masks/00000 contains all the submasks corresponding to train/RGB/00000_0001.png. Additionally, the semantic mask for train/RGB/00000_0001.png is called train/Masks/00000_mask.png. The folder structure can be seen in Figure 9.

Each image and its corresponding masks are $1920 \times 1080$ in resolution. The number of cables, their thickness, color, and bending profile are randomly generated for each image. There are 4 possible colors for the cables - *cyan, white, yellow, black*. The number of cables in each image is randomly sampled from 1 to 10. A sample image from the dataset, along with its corresponding submasks and semantic mask, are portrayed in Figure 10.

## C   Ablation study

The structure of our system is defined by two principal components: the *prompt encoder network* and the *classifier network*, each distinguished by their unique functions and designs. The *prompt encoder network* incorporates a self-attention layer, a sampler attention (cross-attention) layer, a filtering multi-layer perceptron (MLP), and a linear layer for prompt labeling, all critical to its functionality and in their minimal form.

In contrast, the *classifier network*'s architecture is more complex and requires detailed exploration through ablation studies. Our modular approach allows for the independent evaluation of both the classifier and prompt encoder networks. We experimented with multiple configurations of the

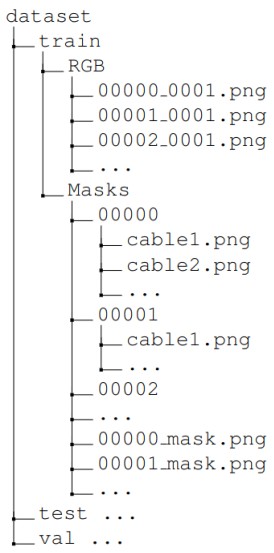

Figure 9: Dataset directory tree

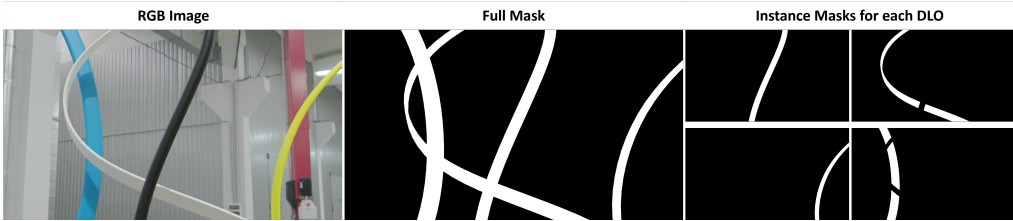

Figure 10: Dataset Example

classifier network to determine the best setup based on classification accuracy on a test dataset. This section will outline the various configurations tested for the classifier network and the resulting labeling accuracies, with a detailed summary provided in Tab. 5.

The *classifier network* is tasked with processing two particular kinds of information: duplication of instance masks and detecting incorrect or low-quality instance masks. These instance masks must undergo a self-attention operation to extract relative information before classification. Initially, we attempted to route these mask tokens through a self-attention layer followed by an MLP classifier. This approach, however, did not converge, as indicated in **A1** in Tab. 5.

Inspired by DETR Carion et al. (2020) and further developed in implementations such as MaskFormer Cheng et al. (2021), we examined the application of learnable token embeddings for classification. Adopting DETR's framework, we initialized N trainable *classifier tokens*. We passed them through a self-attention layer, a cross-attention layer for merging with the mask tokens, and an MLP for classification, achieving a binary classification accuracy of $76.37\%$, documented in **A2** in Tab. 5. A slight modification in this setup, specifically the reordering of the self and cross-attention layers, resulted in an improved accuracy of $81.18\%$, as documented in **A3** in Tab. 5.

Deformable-DETR Zhu et al. (2020) introduced the concept of utilizing localized object queries instead of randomly initialized ones. In **A4**, we substituted the randomly initialized tokens with point prompt embedding directly selected by the prompt encoder without applying any intermediate transformations, leading to an accuracy of $80.26\%$. Furthermore, by applying an MLP transformation to these queries before their integration into the cross-attention layer, as demonstrated in **A5**, we achieved our highest accuracy of $84.83\%$. Consequently, **A5** was selected for all further experiments, showcasing its effectiveness in classification accuracy.

| Config | Queries | Attention Order | Binary Accuracy[%] |
|--------|---------|-----------------|--------------------|
| A1 | Mask Tokens | Only SA | NA(diverged) |
| A2 | Trainable Tokens | SA-CA | 76.37 |
| A3 | Trainable Tokens | CA-SA | 81.18 |
| A4 | Point Prompt Tokens | CA-SA | 80.26 |
| A5 | Point Prompt Tokens (MLP) | CA-SA | **84.83** |

Table 5: Ablations of the classifier network (values in bold signify best performance). SA and CA stand for self-attention and cross-attention respectively. The Keys and Values in the CA are always the mask tokens

## D    Limitations of Foundation Models

The original SAM framework offered three variants based on ViT-B, ViT-L, and ViT-H. We opted the ViT-L model to achieve a balance between computational speed and model performance. Nevertheless, incorporating the ViT-H variant could further enhance our model's performance.

CLIPSeg, designed to effectively manage point prompt embedding, sometimes fails to generate precise heatmap embeddings for specific text prompts, as evidenced in Fig. 11. This issue occasionally impacts the model's performance and its ability to generalize across different scenarios. Looking ahead, we intend to explore the inherent limitations of the backbone models on our setup.

In addition, in the absence of ground truth masks, our model's performance is contingent on the accuracy of the classifier network. In Appendix C, we note that the classifier network currently achieves a peak binary accuracy of $84.83\%$. Future iterations of our model or similar research endeavors will need to focus on enhancing the classifier network to improve the overall model performance..

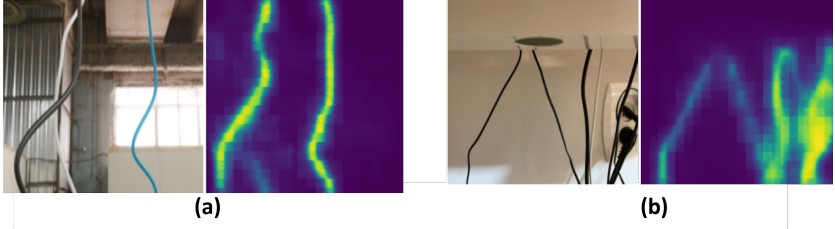

(a)                                                                (b)

Figure 11: CLIPSeg generated heatmaps when using the prompt "cables." Example (a) shows unsuccessful detection of the white cable by CLIPSeg. Example (b) demonstrates a successful detection of all the cables in the frame.

## E    More qualitative results

Figures 12 and 13 show more instance segmentation masks generated by ISCUTE, on datasets from RT-DLO and mBEST as well as on our generated test dataset, respectively. All the images are examples of masks generated **without oracle**.

| ISCUTE (Ours) | RT-DLO Dataset | ISCUTE (Ours) | mBest Dataset |
|---|---|---|---|

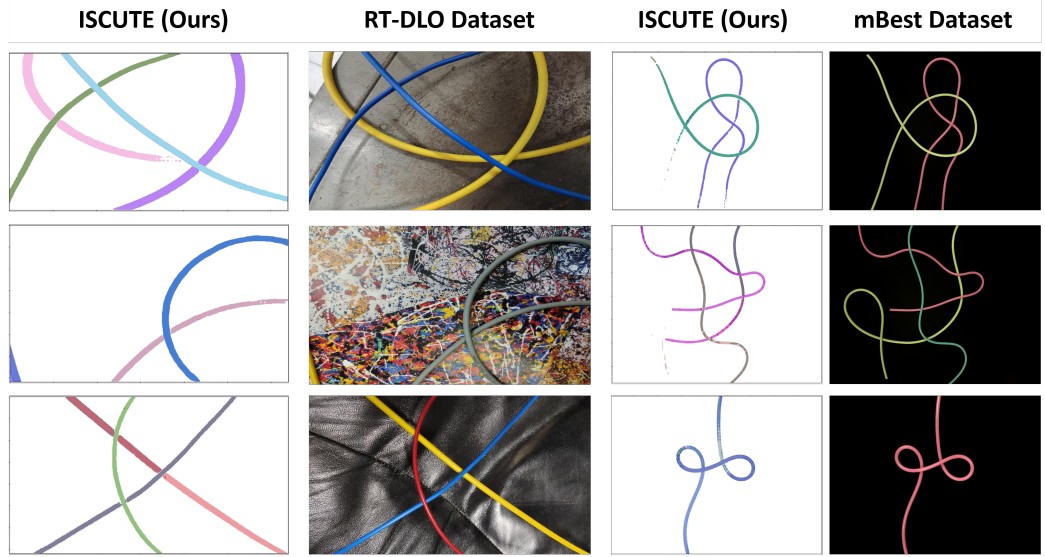

Figure 12: Qualitative results on RT-DLO and mBEST images

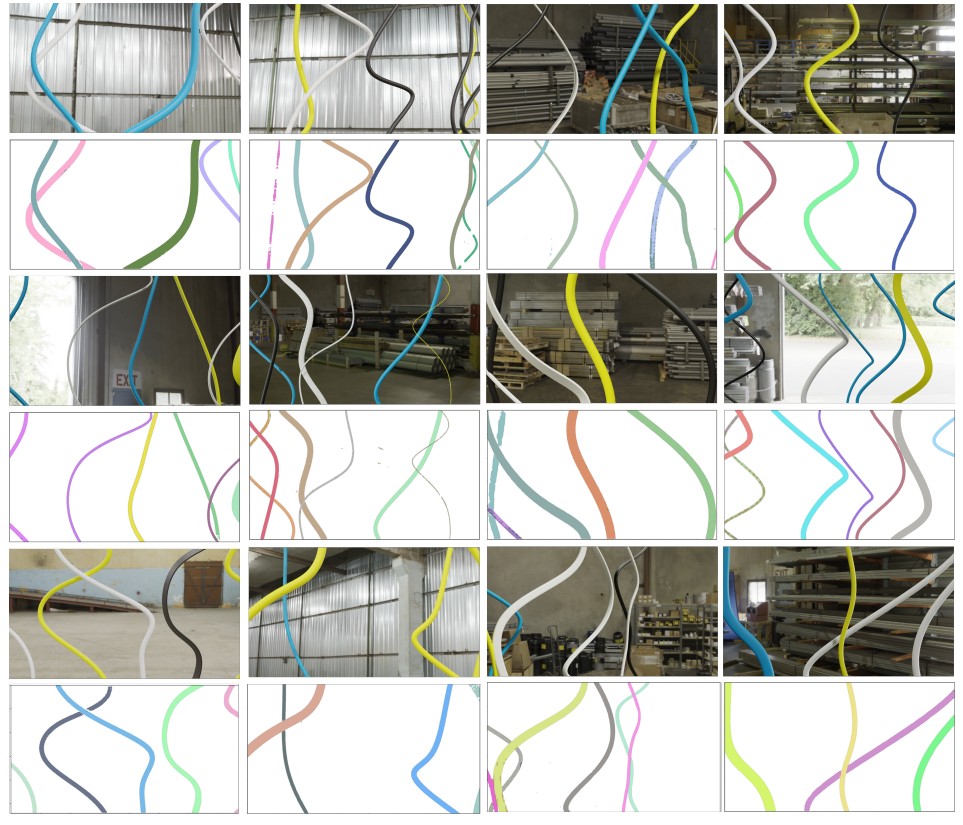

Figure 13: Qualitative results on our test images

