# OpenReview forum: "Adapter to facilitate Foundation Model Communication for DLO Instance Segmentation"
_NeurIPS.cc/2024/Workshop/UniReps — UniReps_

### Official Review · Reviewer_zNeH · 2024-10-03
**Very solid paper**

**Rating:** 8
**Confidence:** 3

**Review:**

### Summary
The authors proposed combining SAM and CLIPSeg by an adapter network for Deformable Linear Object Instance Segmentation. Moreover, the authors created a dataset of industrial cables, using Blender. The proposed approach achieved SOTA results.

### Major strengths:
* The paper is well-organised and easy to follow.
* The proposed approach is simple and effective.
* The authors achieved good results and demonstrated the effectiveness of the zero-shot generalization capabilities of the proposed solution.

### Minor weaknesses:
* The authors mentioned the inference time as one of the limitations of the proposed model. It would be interesting to compare the reference methods.

---

### Official Review · Reviewer_AqXA · 2024-10-04
**The paper proposes a method for DLO Instance Segmentation that combines SAM and CLIPSeg models, showing strong results with two key trainable components. However, its applicability to real datasets and detailed explanation of the training process remain unclear.**

**Rating:** 6
**Confidence:** 3

**Review:**

The paper provides the adaptation technique for Deformable Linear Object (DLO) Instance Segmentation. The method combines the benefits of using SAM and CLIPSeg segmentation techniques to obtain SOTA results on the domain tasks. The architecture uses two trainable components: the prompt encoder network that combines the representations from SAM and CLIPSeg and the Mask classification network that applies segmentation mask filtering. The proposed improvements show great potential for the domain application.

PROS:

The authors show how, smartly, they combine the benefits of two segmentation models to increase the quality of the domain task without training the model.

Both of the adapters are well-justified and supported by experiments.

new dataset for the community.

CONS:

It is not clear why the proposed architecture corresponds with DLO segmentation problem. What type of design components are characteristic of the application? Is it a general domain adaptation technique that can be applied for domain adaption for specific segmentation tasks?

The process of training should be described in more detail.

The experimental evaluation is performed on a blended dataset. It would be interesting to see the results for real datasets as well. The model can be biased towards artificial cases.

---

### Official Review · Reviewer_4TX5 · 2024-10-05
**Accept: A reasonable application of the idea of Latent space communication LSC**

**Rating:** 6
**Confidence:** 5

**Review:**

This work is a development of ClipSeg [8]. A reasonable and well-written paper on how to integrate CLIPSeg models with Segment Anything Models. They keep the models fixed and have adaptors that adapt the feature vectors, and latent representations, and help segment. They get two representations: 1. An embedding heatmap from CLIPSeg 2. Dense Positional Encodings SAM. These 2 are fused together with an attention-based network. There is also a classifier operating on these fused features, which helps with the learning of <not clear>. These fused vectors are then fed into the SAM decoder. The results give reasonable accuracy.

1. In the final draft of the paper, need to cite and draw parallels with ideas on LSC. LSC allows you to use different modules from different pre-trained architectures while keeping them frozen.

https://arxiv.org/abs/2209.15430
https://proceedings.mlr.press/v243/lahner24a.html
https://arxiv.org/pdf/2310.01211

2. Why not expand this work outside the scope of DLO to other domains: cell imagery, generic segmentation datasets, and medical imagery? That way you could do accuracy and computational costs against ClipSeg experiments done in the original ClipSeg paper..

If the authors improve with the above-mentioned suggestions, especially the LSC stuff, it will be a good accept paper for a workshop.

---

### Official Review · Reviewer_2aqd · 2024-10-07
**Adapter to facilitate communication for DLO instance segmentation**

**Rating:** 7
**Confidence:** 2

**Review:**

The paper is well written and explained. The authors propose a framework that (i)embeds a text prompt to point prompt embeddings and (ii)mask network to eliminate low quality data and (iii)A new dataset.

---

### Decision · Program_Chairs · 2024-10-10

**Decision:**

Accept

**Comment:**

In light of the positive reviewers' feedback and relevancy of the submission, we are pleased to accept this paper for presentation at UniReps 2024. We kindly ask the authors to incorporate the reviewers' suggestions and feedback in the final camera-ready version of the manuscript.